# Brain morphology in Anorexia Nervosa and its subtypes: A multi-cohort study of individual participant data

Fabio Bernardoni[1*], Dominic Arold[1], Luis Schoppik[1], Klaas Bahnsen[1,2], Ruiyang Ge[3], Clara Moreau[4], Lasse Bang[5], Federico D'Agata[6], Giovanni Abbate-Daga[6,7], Christian K. Tamnes[8,9], Iain Campbell[10], Owen O'Daly[11], Ulrike Schmidt[11], Guido Frank[12,13], Stefanie Horndasch[14,15], Andreas Hess[16,17,18], Arnd Dörfler[19], Hans-Christoph Friederich[20], Joe Simon[20], Angela Favaro[21], Luca Lavagnino[22], Christina E. Wierenga[12,13], Amanda Bischoff-Grethe[12,13], Amy E. Miles[23], Allan Kaplan[23], Aristotle Voineskos[23], Paul A. M. Smeets[24,25], Annemarie A. van Elburg[26,27], Unna Danner[26,27], Sophia I. Thomopoulos[28], Laura Berner[29], Neda Jahanshad[28], Sophia Frangou[3,29], Joseph A. King[1], Paul Thompson[28], Stefan Ehrlich[1,30]

1 Translational Developmental Neuroscience Section, Division of Psychological and Social Medicine and Developmental Neurosciences, Faculty of Medicine, Technische Universität Dresden, Dresden, Germany, 2 Maurice Wohl Clinical Neuroscience Institute, Department of Psychological Medicine, Institute of Psychiatry, Psychology and Neuroscience, King's College London, London, United Kingdom, 3 Djavad Mowafaghian Centre for Brain Health, University of British Columbia, Vancouver, British Columbia, Canada, 4 Centre de recherche CHU Sainte Justine, Department of Psychiatry and Addictology, University of Montreal, Montreal, Québec, Canada, 5 Department of Child Health and Development, Norwegian Institute of Public Health, Oslo, Norway, 6 Department of Neurosciences 'Rita Levi Montalcini', University of Turin, Turin, Italy, 7 Eating Disorders Center for Treatment and Research, University of Turin, Turin, Italy, 8 PROMENTA Research Center, Department of Psychology, University of Oslo, Oslo, Norway, 9 Division of Mental Health and Substance Abuse, Diakonhjemmet Hospital, Oslo, Norway, 10 Centre for Research in Eating and Weight Disorders, Institute of Psychitry, Psychology and Neuroscience, King's College London, London, United Kingdom 11 Department of Neuroimaging, Institute of Psychiatry, Psychology and Neuroscience, King's College London, London, United Kingdom, 12 Department of Psychiatry, University of California San Diego, La Jolla, California, United States of America, 13 Eating Disorders Center for Treatment and Research, University of California San Diego, La Jolla, California, United States of America, 14 Department of Child and Adolescent Psychiatry, Bielefeld University, Medical School and University Medical Center OWL, Protestant Hospital of the Bethel Foundation, Bielefeld, Germany, 15 Department of Child and Adolescent Psychiatry, University Clinic Erlangen, Erlangen, Germany, 16 Institute of Experimental and Clinical Pharmacology and Toxicology, Emil Fischer Center, University of Erlangen-Nuremberg, Erlangen, Germany, 17 Department of Neuroradiology, University of Erlangen-Nuremberg, Erlangen, Germany, 18 FAU NeW - Research Center for New Bioactive Compounds, Friedrich-Alexander-Universität Erlangen-Nürnberg, Erlangen, Germany, 19 Department of Neuroradiology, University of Erlangen-Nuremberg, Erlangen, Germany, 20 Centre for Psychosocial Medicine, Department of General Internal Medicine and Psychosomatics, University Hospital Heidelberg, Heidelberg, Germany, 21 Padova Neuroscience Center, Department of Neurosciences, University of Padova, Padova, Italy 22 Department of Psychiatry and Behavioral Sciences, University of Texas Health Science Center, Houston, Texas, United States of America, 23 Campbell Family Mental Health Research Institute, Centre for Addiction and Mental Health, Toronto, Ontario, Canada, 24 UMC Utrecht Brain Center, Utrecht University, Utrecht, the Netherlands, 25 Division of Human Nutrition and Health, Wageningen University, Wageningen, the Netherlands, 26 Altrecht Eating Disorders Rintveld, Altrecht Mental Health Institute, Zeist, the Netherlands, 27 Faculty of Social Sciences, Utrecht University, Utrecht, the Netherlands, 28 Imaging Genetics Center, Stevens Institute for Neuroimaging and Informatics, Keck USC School of Medicine, Marina del Rey, California, United States of America, 29 Department of Psychiatry, Icahn School of Medicine at Mount Sinai, New York, New York, United States of America, 30 Eating Disorders Research and Treatment Center, Department of Child and Adolescent Psychiatry, Faculty of Medicine, Dresden University of Technology, Dresden, Germany

* fabio.bernardoni@ukdd.de

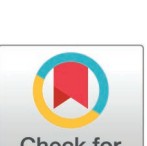

purpose. The work is made available under the Creative Commons CC0 public domain dedication.

**Data availability statement:** Individual-level data underlying the findings of this study cannot be shared publicly because they are governed by site-specific ethical approvals and national and institutional data protection regulations at the 11 contributing ENIGMA Eating Disorders Working Group sites. The study authors are not the legal custodians of these data. Data access inquiries must be directed to the relevant institutional data custodian or ethics/governance office at each contributing site (Denver: COMIRB@ucdenver.edu, Dresden: ethikkommission@mailbox.tu-dresden.de, Erlangen: ethik-kommission@fau.de, Heidelberg: ethikkommission-I@med.uni-heidelberg.de, London: hampstead.rec@hra.nhs.uk, rec@kcl.ac.uk, Oslo: rek-sorost@medisin.uio.no, Padova: comitato.etico@aopd.veneto.it, San Diego: irb@ucsd.edu, Torino: comitatoetico@pec.cittadellasalute.to.it, Toronto: research.ethics@camh.ca, Utrecht: metc@nedmec.nl, researchoffice@umcutrecht.nl). Any access is subject to local approval procedures and applicable legal and contractual restrictions. Summary-level data underlying the reported findings are provided in Tables A and B in the S1 Appendix. The code used for the analyses is available on OSF (https://osf.io/xrjkf/overview?view_only=b73a380dfaf94c36a-69d9354e0c92679) or through the DOI (https://doi.org/10.17605/OSF.IO/XRJKF).

**Funding:** This work was supported by the Else Kröner-Fresenius-Stiftung (https://ekfs.de/, Grant No. 2019_A118 [to FB, DA, and LS]), the European Union and the Saxon State Parliament (https://www.era-learn.eu/network-information/networks/personalised-medicine, EP PerMed project BIOREXIA; 100770101 [to SE]), German Research Foundation (https://www.dfg.de/en, Grant Nos. SI 2087/2-1 and BR 4852/1-1 [to JJS] and Grant Nos. EH 367/5-1 and EH 367/7-1 [to SE]), the Carus Promotionskolleg from the Medizinische Fakultät Carl Gustav Carus, Technische Universität Dresden (https://tu-dresden.de/med/mf/forschung-internatio-nales/nachwuchsfoerderung-dscs/internefoerd-erprogramme/carus-promotionskolleg-dresden, to KB), National Institutes of Health (https://www.nih.gov/, Grant No. R21MH86017 [to AB-G and CEW], Grant No. R01MH113588 [to AB-G, and CEW], Grant Nos. K23MH080135 and R01MH096777 [to GKWF and MES]),

# Abstract

## Background

In a recent coordinated meta-analysis of neuroimaging data, we reported gray matter (GM) alterations in acutely underweight patients with anorexia nervosa (AN). Here, we extend these findings by examining individual variation in brain structure within AN, individual-level differentiation between AN and healthy controls (HC), and differences between AN subtypes, with potential relevance for understanding clinical heterogeneity.

## Methods and findings

We analyzed individual-level data from 11 international sites in the ENIGMA Eating Disorders Working Group, including 570 female participants with AN and 739 HC. We examined cortical thickness, cortical surface area and subcortical volumes in AN versus HC using three complementary approaches: (i) group-level differences in a mega-analysis correcting for age effects, (ii) frequencies of extreme deviations (infra-/supranormal; $z < -1.96$/$z > 1.96$) based on normative reference models by the CentileBrain Initiative, and (iii) individual-level classification performance using machine learning. The same analytic framework was applied to compare AN restricting versus binge-eating/purging subtype, additionally correcting for BMI effects.
Mega-analyses reinforced previous meta-analytic findings of pronounced and wide-spread GM deficits in AN compared to HC. Normative modelling revealed that the frequency of infranormal z-scores (23/68 cortical thickness, 13/14 subcortical volume metrics) and supranormal z-scores (35/68 cortical thickness, 17/68 cortical surface area metrics) was significantly higher in AN than expected based on reference data. Individuals with AN could be reliably differentiated from HC using machine-learning classifiers (ROC–AUC = 0.75–0.81). In contrast, neither group-level differences nor frequency of extreme z-scores differed between AN subtypes, and individuals with different subtypes could not be reliably differentiated from each other. Importantly, the observational design cannot distinguish neurobiological differences related to AN from the effects of starvation or low BMI in the AN versus HC analyses. The lack of differences between subtypes does not exclude brain structural differences between AN subtypes that might be detectable with other modalities or analytic approaches.

## Conclusion

Using a mega-analytic approach, we confirm widespread GM deficits in AN, show that these alterations are (in some patients) extreme, and demonstrate that they enable robust classification with superior performance compared to most MRI-based psychiatric classification studies. The absence of differences between AN subtypes may reflect shared neurobiology, though other imaging modalities may reveal distinctions beyond brain structure.

Swiss Anorexia Nervosa Foundation (https://www.fundraiso.com/en/organisations/schweizerische-anorexia-nervosa-stiftung, Project No. 57-16 [to JJS] and to SE), National Institute for Health Research Mental Health Biomedical Research Centre at the South London (https://www.kcl.ac.uk/ioppn/research/nihr-maudsley-brc) and Maudsley NHS Foundation Trust and King's College London (https://www.kcl.ac.uk/ioppn/our-connections/slam, ICC, OO, and UHS), Research Council of Norway (https://www.forskningsradet.no/en/, Grant Nos. 288083 and 323951 [to CKT]), South-Eastern Norway Regional Health Authority (https://www.helse-sorost.no/, Grant Nos. 2021070, 2023012, and 500189 [to CKT]), National Institute of Health Research Senior Investigator Award (https://www.nihr.ac.uk/, to UHS), CAMH AFP Innovation Fund (https://www.camh.ca/, CAM-14-001 [to AM, AK, and AV]), and Technische Universität Dresden SFB 940 (https://tu-dresden.de/, to SE). The ENIGMA ED Working Group acknowledges the National Institutes of Health Big Data to Knowledge award for foundational support and consortium development (https://commonfund.nih.gov/bd2k, Grant No. U54 EB020403 [to PMT]). The normative CentileBrain Models were developed with support by the National Institutes of Mental Health (R01MH134962 [to SF and PMT]). The funders had no role in study design, data collection and analysis, decision to publish, or preparation of the manuscript.

**Competing interests:** I have read the journal's policy and the authors of this manuscript have the following competing interests: LAB is a scientific advisor to Juniver, LLC. The other authors report no conflict of interest.

**Abbreviations:** AN, anorexia nervosa; BMI, body mass index; CT, cortical thickness; DSM-IV or DSM-5, Diagnostic and Statistical Manual of Mental Disorders, Fourth or Fifth Edition; ED, Eating Disorders; FDR, false discovery rate; GLMs, General Linear Models; GM, gray matter; HC, healthy controls; ICV, intracranial volume; ICD-10, International Classification of Diseases, Tenth Revision; MRI, magnetic resonance imaging; PCA, principal component analysis; PR-AUC, precision-recall area under the curve; ROC-AUC, area under the receiver operator characteristic curve; SA, surface areas; STROBE, Strengthening the Reporting of Observational Studies in Epidemiology; SV, subcortical volumes; SVM, Support Vector Machine.

## Author summary

### Why was this study done?

- Previous large-scale studies have shown that people with anorexia nervosa (AN) who are currently underweight often have reduced cerebral gray matter, but it was unclear to what extent these changes vary across individuals.

- We also aimed to determine whether brain scans can reliably distinguish individuals with AN from healthy people, and whether brain structure differs between AN subtypes: restricting type versus binge-eating/purging type.

### What did the researchers do and find?

- We combined structural brain MRI data from 11 international sites, analyzing 570 females with AN and 739 healthy controls.

- Brain structure was examined at multiple levels: average group differences, individual deviations relative to a large normative reference dataset, and machine-learning classification.

- On average, we confirmed that people with AN have reduced gray matter compared with healthy controls (90% of cortical thickness, 84% of cortical surface area, and 100% of subcortical volume metrics affected). However, we also observed a higher-than-expected number of "extreme" brain measurements and overall greater variability, indicating that brain structure changes in AN are highly heterogeneous across individuals.

- Despite this heterogeneity, a multivariate pattern across many brain features reliably distinguished AN from healthy controls (ROC–AUC ~0.75–0.81), while AN subtypes could not be reliably differentiated.

### What do these findings mean?

- Brain structure changes in acute AN do not reflect a single uniform pattern; instead, there is marked biological heterogeneity, with some individuals showing pronounced alterations while others are relatively less affected.

- Nevertheless, the overall *pattern across multiple brain features* is sufficiently consistent to distinguish AN from healthy controls, with better performance than is typical for MRI-based classification studies in psychiatry.

- The absence of structural differences between AN subtypes suggests shared neurobiology in brain morphology. However, the study is limited by its observational design, focus on currently underweight females, and inability to determine whether brain differences are causes or consequences of illness or how they evolve with recovery.

## Introduction

Anorexia nervosa (AN) is an eating disorder characterized by severe dietary restriction resulting in low weight and a high mortality rate due to complications of starvation [1]. While the exact aetiology remains unclear, its biological underpinnings are widely acknowledged [2]. No specific pharmacological treatment exists for AN, with therapy typically focusing on psychotherapy and weight restoration [3]. However, clinical presentation and outcomes are highly variable—a significant proportion of patients experience relapses and chronicity, and even suicide or premature death [1]. Understanding why standard therapies work for some patients but not others is critical. While several factors influence clinical trajectories in AN [4], this study specifically focuses on individual-level variation in brain structure, building on prior evidence that such neurobiological differences can help predict treatment outcomes [5].

Previously, a prospective harmonized meta-analysis from the ENIGMA (Enhancing NeuroImaging Genetics through Meta-Analysis) Eating Disorders (ED) Working Group (http://enigma.ini.usc.edu/ongoing/enigma-eating-disorders/) revealed sizeable and widespread gray matter (GM) reductions associated with low weight and body mass index (BMI) in AN [6]. Evidence from related populations suggests that GM reductions may also occur in the context of low BMI outside AN [7], including in early-onset restrictive eating disorders in very young children [8] and in population-based samples of underweight preadolescents (e.g., Generation R cohort [9]). Going beyond these findings, we collected the largest multi-site cohort of individual-level data within the ENIGMA ED Working Group for participants with AN and healthy controls (HC). In contrast to our previous group-level meta-analysis, analyzing individual-level data enables examination of variability related to clinical AN subtypes [10]. Clinically, the restricting subtype (AN-R) is characterized by severe food restriction, which may occur with or without increased energy expenditure (e.g., excessive exercise), while the binge-eating/purging subtype (AN-BP) is marked by purging compensatory behaviors (e.g., self-induced vomiting, laxative misuse, and enemas) alongside food restriction [10]. Importantly, AN-BP is distinguished from bulimia nervosa by the persistence of significantly low body weight, despite the presence of binge-eating and purging behaviors [10]. Although diagnostic crossover between AN-R and AN-BP is common, particularly with increasing illness duration [11], and the validity of this subtype distinction has been debated [12,13], these subtypes are currently retained in diagnostic systems to describe differences in clinical presentation. AN-R is often associated with an earlier age of onset, predominance in adolescent samples, and a more stable course [14]. AN-BP becomes more prevalent with longer illness duration, has a more fluctuating course, higher levels of suicidality [15–17], relapse [11,18], and co-occurring psychiatric symptoms [11,19–22]. While no differences in cerebral blood flow between these subtypes have been reported previously [23], the large sample size of the present study provides a robust opportunity to investigate potential differences in brain morphology. This may advance understanding of the biological underpinnings of structural alterations in AN, inform the ongoing debate regarding the validity of the current subtype distinction [13], and ultimately support the development of more targeted treatments [5]. Capitalizing on the richness of individual-level data, we adopted three complementary approaches to more precisely characterize brain alterations associated with AN and AN-subtypes at the group and individual level.

First, to examine group-level differences, we conducted a mega-analysis, a multisite data analysis where the individual-level data are shared rather than just the summary statistics from each site. This offers several advantages. For example, research conducted by various ENIGMA working groups has shown that mega-analyses yield lower standard errors and narrower confidence intervals compared to meta-analyses [24–27]. These improvements become even more pronounced when adjusting for different scanning devices and sequences using the ComBat method [28], as opposed to the random-effects approach typically used in meta-analyses [29].

Second, given the clinical heterogeneity observed among individuals with the same psychiatric diagnosis [30,31], which has been linked to variability in the distribution of extreme structural magnetic resonance imaging (MRI) values [32–35], we examined extreme z-scores (infra- or supra-normal) in regional cortical and subcortical brain morphometry in patients with AN relative to a normative reference sample. The normative reference was defined using models from the Centile-Brain Initiative [36], which estimate expected values and normative ranges for brain phenotypes (e.g., cortical thickness

(CT) in a given brain region), based on age, sex and global GM variables. The location and frequency of infra- and supra-normal z-scores were then compared between AN and the normative reference, and between AN-BP and AN-R.

Third, differences between individuals with AN and HC, or between individuals with AN-R and AN-BP might be multivariate, characterized by complex, nonlinear patterns involving multiple metrics of brain structure (e.g., CT in different regions), rather than univariate differences, which focus on single measures such as CT for a specific cortical region. To examine multivariate differences in structural MRI images, we applied machine learning to classify individuals with AN from HCs, as well as individuals with AN-R from those with AN-BP. The classification performance indicates the extent to which measurable and consistent differences in brain structure can predict group membership. By analyzing the feature importances from the model, we can further identify the specific brain regions where these multivariate differences are located.

Together, these complementary analyses were designed to address the overarching question of whether AN is associated with consistent alterations in brain morphology at both the group and individual level, and whether these alterations differ between AN-R and AN-BP. Based on previous structural MRI findings in AN, we hypothesized that individuals with AN would show lower brain morphometric values than HCs [6], particularly for CT and SV. We further hypothesized that individual-level normative deviations would be more frequent in AN than in the normative reference sample, reflecting heterogeneity in the spatial distribution of extreme morphometric values. Finally, based on a previous single-site study [5], we expected that multivariate models would classify AN versus HC above chance level, indicating distributed structural differences in brain morphology. Because previous structural MRI studies have rarely examined morphological differences between AN-R and AN-BP directly, and because available evidence does not provide a consistent basis for predicting the direction or regional distribution of such differences, we did not formulate a directional hypothesis for subtype comparisons.

## Methods

### Study samples

Thirteen cohorts contributed individual-level data to the AN arm of the ENIGMA ED working group, 12 of which included subtype information for each participant with AN. Inclusion and exclusion criteria were as in our previous study [6]. Specifically, patients with AN and HC were selected within each site according to standardized inclusion and exclusion criteria, as in the main analysis by Walton and colleagues [6], but patients were not stratified into two groups according to their weight status. Compared to the sample in Walton and colleagues [6], we excluded cohorts that could not share single participant data, while for some sites, data for more participants became available. To be included in the AN group, participants had to be female and meet the DSM-IV-TR, DSM-5, or ICD-11 criteria for AN, i.e., restriction of energy intake leading to significantly low body weight, intense fear of gaining weight (or persistent behavior interfering with weight gain), and disturbance in body weight/shape experience. Specifically, participants with AN had a BMI < 18.5 kg/m² for adults (>18 years old) or below the 10th percentile for age-adjusted BMI in adolescents. HC participants were also female, with BMI > 17.5 kg/m² for adults or above the 10th percentile for age-adjusted BMI in adolescents, and with no current or lifetime diagnosis of any eating disorder. Although this range includes some underweight individuals, all HC were screened as healthy eaters without evidence of an eating disorder or somatic causes of low weight. Our rationale for including adult HC with BMI < 18.5 kg/m² (12/449 ~ 2.7%) was to preserve a control group representative of the general population, in which a small proportion of individuals have BMI < 18.5 kg/m² without an eating disorder. We included only female participants because AN has a markedly higher prevalence in females [37], and restricting the sample to females reduced heterogeneity and avoided confounding by sex-related differences in brain morphology. For all participants, the following exclusion criteria were applied: current comorbid severe psychiatric disorders (such as schizophrenia/schizoaffective, bipolar disorder, and substance dependence), current severe neurological disorders, significant current or chronic medical illness that can explain most of the weight loss, significant lifetime neurological illness/accidents that may have affected the brain, preterm birth <30 weeks, and/or developmental disorders.

**Case-control sample (AN versus HC).** We aggregated data from 13 cohorts from 11 sites with a combined sample size of $n = 570$ patients with AN and $n = 739$ HC. The age in AN was on average 20.54 (6.44) years and in HC 21.10 (5.79) years, while BMI in AN was 15.49 (1.54) kg/m$^2$ and in HC 21.33 (2.20) kg/m$^2$. Participants were matched within each sample according to age. Further information on how age and BMI were distributed across sites is reported in Table A and Figs A, B in S1 Appendix.

**Sample of cases with subtype information (AN-R versus AN-BP).** In 12 cohorts from 10 sites where subtype information was available for patients with AN, $N = 417$ were AN-R and $N = 135$ were AN-BP. AN subtype was assessed at each site using standardized diagnostic interviews or expert clinical evaluations based on the Diagnostic and Statistical Manual of Mental Disorders, Fourth or Fifth Edition (DSM-IV or DSM-5), or the International Classification of Diseases, Tenth Revision (ICD-10) (Section A.3 in S1 Appendix). Further information on how subtype, age, and BMI were distributed across sites is reported in Table B, and Figs C and D in S1 Appendix.

### Ethics statement

All participating cohorts obtained approval from their local institutional review boards or ethics committees, and all study participants, or their legal guardians where applicable, provided written informed consent. Ethical approval was granted by the Colorado Multiple Institutional Review Board; the Ethics Commission at Dresden Technical University; the Ethics Committee of the University Hospital of Erlangen; the medical ethics committee of the Medical Faculty Heidelberg at Ruprecht-Karls-University Heidelberg; the London—City Road and Hampstead Research Ethics Committee; the King's College London Psychiatry, Nursing and Midwifery Research Ethics Subcommittee; the Norwegian Regional Committee for Medical and Health Research Ethics; the ethics committee of Padova Hospital; the University of California, San Diego Human Research Protections Program; the Comitato Etico Interaziendale A.O.U. Città della Salute e della Scienza di Torino—A.O. Ordine Mauriziano—A.S.L. Città di Torino; the Centre for Addiction and Mental Health Research Ethics Board; and the Institutional Review Board of the University Medical Center Utrecht. Site-specific protocol numbers and additional details are provided in Section A.1 in S1 Appendix.

### Image acquisition and processing

As in our previous study [6], all sites processed $T_1$-weighted structural brain scans using FreeSurfer (http://surfer.nmr.mgh.harvard.edu) and extracted subcortical volumes (SV) for eight regions, CT and surface areas (SA) for 34 Desikan–Killiany atlas regions (per hemisphere), mean CT (per hemisphere), and total SA [38]. $T_1$-weighted images were acquired across sites using MRI scanners from General Electric Healthcare, Siemens, and Philips, including models such as Signa, Trio/Magnetom Trio, Skyra, Avanto, Achieva, MR750, and Echospeed, operating at field strengths of 1.5T or 3T (see Table C in S1 Appendix for site-specific details). The ENIGMA protocol for quality assurance was performed for each site prior to analysis, and included visual checks of the cortical segmentations and region-by-region removal of values for segmentations found to be incorrect (http://enigma.usc.edu/protocols/imaging-protocols). Histograms of all regions' values for each site were also produced for visual inspection.

### Group comparisons

Our univariate mega-analyses of brain structural data aimed to (i) reproduce group differences between AN and HC from our previous meta-analysis [6] and (ii) examine group differences between AN subtypes (restricting versus binge-eating/purging), which was not possible in the study by Walton and colleagues [6], with the goal of identifying subtype-specific effects independent of illness severity.

We used ComBat with generalized additive models (ComBat-GAM) [39] to remove site effects from brain structural data while preserving biological variability from factors of interest and confounds (e.g., age). Specifically, following Pomponio

and colleagues [39], ComBat-GAM was used to retain nonlinear effects of age, linear effects of BMI, linear effects of intra-cranial volume (ICV, if applicable), and subtype/diagnosis dependences. Subsequently, we used General Linear Models (GLMs) to examine group differences while adjusting for relevant covariates. Age (modelled with linear and quadratic terms) was included in all analyses and for all metrics in both the AN versus HC and AN-R versus AN-BP comparisons. BMI was included as a covariate only in the AN-R versus AN-BP analyses, where it was considered a proxy for illness severity or recovery stage rather than an inherent subtype characteristic [40,41]. ICV was included for regional surface area metrics and subcortical volumes in both the AN versus HC and AN-R versus AN-BP comparisons. To account for multiple comparisons, statistical significance was set at a false discovery rate (FDR) $q < 0.05$ [42].

## Normative modeling

Normative morphometry models for regional CT, SA, and SV were provided by CentileBrain, a major initiative of the Lifespan Working Group of the ENIGMA Consortium [36]. The CentileBrain brain morphometry models (available at https://centilebrain.org/) are empirically validated sex-specific normative models of FreeSurfer-extracted regional metrics of SV, CT, and SA derived from an ethnoracial diverse sample of >37,000 healthy individuals, aged 5–90 years [36]. For each individual regional morphometric measure, the models provide a z-score which quantifies the deviation from the population mean.

In this study, the CentileBrain female-specific models of regional CT and SA and SV were applied to the corresponding harmonized data of the sample of patients with AN. Following Haas and colleagues [43], we defined regional "extreme" z-scores as infranormal if $z < -1.96$ or supranormal if $z > 1.96$, corresponding to the 2.5th and 97.5th percentiles, respectively. Intermediate values (i.e., $-1.96 < z < 1.96$) were designated as within "normal" range. For each metric and AN group (AN, AN-R, and AN-BP), we determined the percentage of infra-/supra-normal z-scores that would be significantly higher compared to the normative reference after FDR correction (Section A.6 in S1 Appendix). Furthermore, we examined differences between AN-R and AN-BP in the proportion of individuals with supra- or infra-normal z-scores using two-proportion z-tests, with FDR correction applied to account for multiple comparisons [42]. To interpret group differences, we computed the standard deviation of z-scores across individuals for each metric and averaged these values to obtain global CT, SA, and SV heterogeneity indices. Statistical significance was evaluated using a bootstrap approach (Section A.6 in S1 Appendix).

## Machine learning classification

**Classification pipelines.** We trained classifiers to distinguish between AN and HC based on all metrics used in the mega-analysis, as well as between AN-R and AN-BP. The analysis followed the approach introduced by Arold and colleagues [5]. However, here we included an additional preprocessing step to harmonize multisite data using ComBat-GAM, as in the mega-analysis [39]. To prevent information leakage, ComBat-GAM and all preprocessing steps were embedded within the same nested 10-fold cross-validation scheme used for hyperparameter optimization and performance estimation [44]. Specifically, for each cross-validation iteration, ComBat-GAM parameters were estimated exclusively on the training folds and then applied to both the training and corresponding held-out test fold. Hyperparameter optimization was performed using grid search within the training data only, and model performance was evaluated on the unseen test fold. All preprocessing, model training, and evaluation steps were implemented within a single Scikit-learn pipeline using version 1.3.3 in Python 3.8.10 [45].

Following Arold and colleagues [5], after Combat-GAM harmonization, we regressed out linear effects of covariates from brain structural metrics [46]. Specifically, we adjusted for age in all models, BMI in the AN-R versus AN-BP classification to reduce confounding by illness severity, and ICV for SA and SV metrics in both classification tasks. This adjustment aimed to enforce the classifier to take decisions based on brain structural metrics themselves, rather than imbalances between groups in covariates with an effect on brain structure. Subsequent steps were principal component analysis (PCA) for dimensionality reduction, and a linear Support Vector Machine (SVM) or a neural network (to evaluate whether nonlinear patterns in the data could enhance performance) for classification (Section A.7 in S1 Appendix). Both the SVM

and neural network produce a continuous machine learning–based risk score, which can be thresholded to yield a binary prediction.

**Hyperparameters optimization and performance estimation.** We jointly optimized the model hyperparameters (e.g., number of PCA components) via a stratified (i.e., keeping the group proportions constant across partitions) 10-fold cross-validated grid search (Table D in S1 Appendix).

Since the classes (AN-R versus AN-BP and AN versus HC) were imbalanced within and across sites (Tables A, B, Figs A, C in S1 Appendix), we used precision-recall area under the curve (PR-AUC) as optimization and performance metric [47] and weighted the cost of misclassification in the classifier loss function by the inverse of the group frequency. We also report the area under the receiver operator characteristic curve (ROC-AUC) to compare classifiers (AN/HC versus AN-R/AN-BP) trained on data sets with different class ratios. Since our optimization procedure involved the optimization of hyperparameters (Section A.8, Table D in S1 Appendix), to obtain unbiased model performance estimates, we applied (to the whole model pipeline) nested cross-validation (Section A.9 in S1 Appendix). Subsequently, permutation tests were performed to establish whether the obtained performance was significantly above chance (Section A.10 in S1 Appendix), thus suggesting the presence of multivariate group differences.

Although cross-validation applied to pipelines employing ComBat-GAM provides unbiased performance estimates for sites included in the training data, this performance might not replicate when applied to completely new data from sites not represented in the original dataset. To better approximate such scenarios, we also used Leave-Site(s)-Out cross-validation (LSsO, [48]). In this approach, validation and testing were conducted using data from sites not used for training (Section A.11 in S1 Appendix), and classification pipelines included linear confound regression, PCA and the SVM, but not ComBat-GAM (Section A.7 in S1 Appendix). Finally, for classifiers performing above chance, we assessed the impact of confounding variables (e.g., age, ICV) on performance estimates a posteriori (Section A.12 in S1 Appendix), as recommended by Dinga and colleagues [49].

**Explainable Artificial Intelligence.** If the performance of a classifier was significantly better than chance, we estimated the importance of each feature for the prediction, i.e., how much information useful to differentiate AN from HC or AN-R from AN-BP it contained. We applied the method suggested by Haufe and colleagues [50]. A positive/negative feature importance indicates whether a higher/lower feature value was characteristic for AN (Section A.13 in S1 Appendix).

This study is reported as per the Strengthening the Reporting of Observational Studies in Epidemiology (STROBE) guideline (S1 Checklist).

## Results

### Demographics

Descriptive statistics per site are reported in Tables A, B in S1 Appendix. As expected, within all cohorts and in the pooled sample, BMI was lower in AN compared to HC ($p < .001$, see Fig B in S1 Appendix). When comparing AN subtypes, BMI was lower in AN-R compared to AN-BP in some cohorts (Padova, Denver, Heidelberg, Dresden, see Fig D in S1 Appendix) and overall (pooled sample: $p < .001$). Age was lower in patients with AN compared to HC in some cohorts (Denver, Dresden, see Fig B in S1 Appendix), and overall ($p = .001$), and it was lower in patients with AN-R compared to those with AN-BP in the same cohorts (Denver, Dresden, see Fig D in S1 Appendix) and overall ($p < .001$). The age difference in subtypes may reflect the more typical transition from the AN-R to AN-BP over time [19,51,52].

### Univariate comparisons

In this mega-analysis, regional metrics of CT, SA, and SV were lower in most regions in AN compared to HC. CT was lower in 61 regions (Fig 1a), with the largest effects in the left superior ($d = −0.96$, 95% CI [−1.08, −0.85]; $p < .001$) and inferior ($d = −0.96$, 95% CI [−1.07, −0.84]; $p < .001$) parietal cortex. Mean(SD) effect size across these regions was

PLOS Medicine

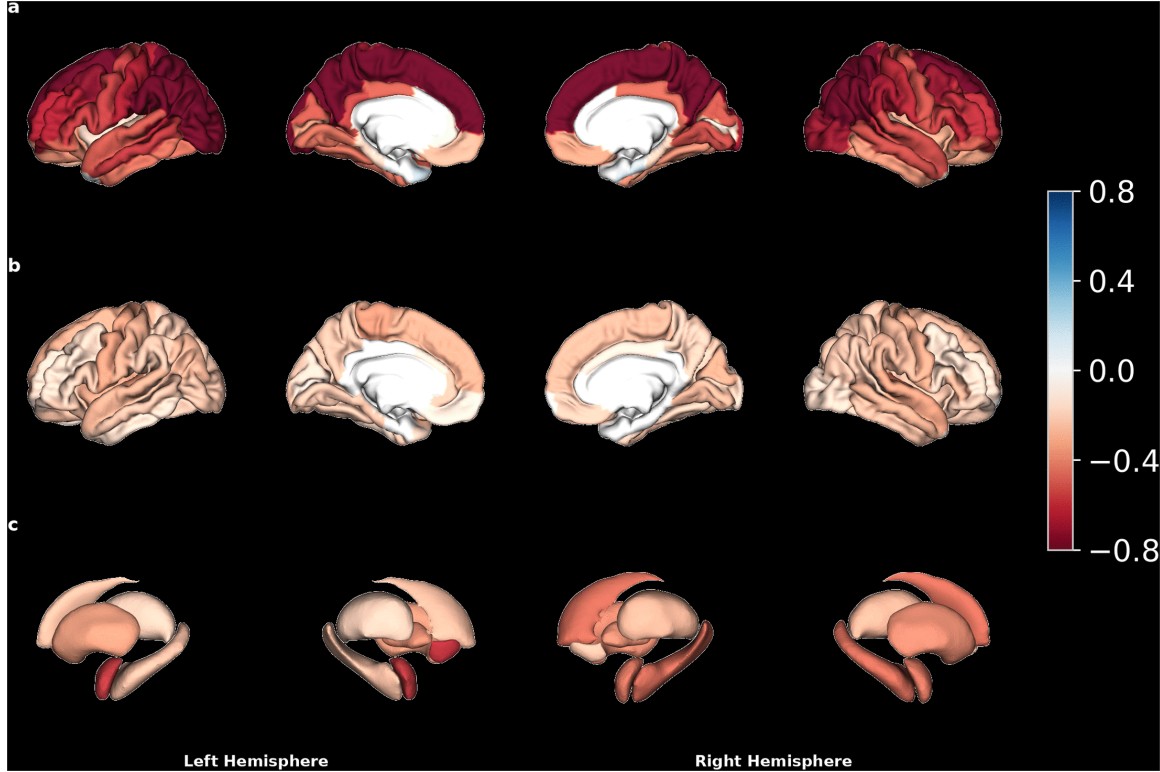

**Fig 1. Effect sizes for anorexia nervosa (AN) vs. healthy control (HC) group differences in brain structure.** Panels show Cohen's d values for group differences in **(a)** cortical thickness (CT), **(b)** cortical surface area (SA), and **(c)** subcortical volume (SV). Effect sizes were estimated from general linear models including age as a covariate, with up to quadratic age effects. Cohen's d is coded as AN minus HC; therefore, negative values indicate lower values in the AN group compared with the HC group, whereas positive values indicate higher values in the AN group. Regions are coloured only when the corresponding group difference survived false discovery rate (FDR) correction at $q < .05$. Red indicates significantly lower values in AN, and blue indicates significantly higher values in AN. White regions indicate nonsignificant group differences after FDR correction. Colour intensity reflects the magnitude of the effect size, as shown by the colour bar.

d = −0.54(0.22). SA was lower in 57 regions (Fig 1b), with the largest effects in the right and left transverse temporal gyrus (d = −0.45, 95% CI [−0.56, −0.34]; $p < .001$ and d = −0.36, 95% CI [−0.47, −0.25]; $p < .001$). Mean(SD) effect size across these regions was d = −0.23(0.07). All SV were lower (except the ventricles which were enlarged, Fig 1c), with the largest effects in the left and right thalamus (d = −0.61, 95% CI [−0.72, −0.49]; $p < .001$ and d = −0.58, 95% CI [−1.07, −0.84]; $p < .001$). Mean(SD) effect size across these regions was d = −0.37(0.13). CT was higher in the left temporal pole (d = 0.15, 95% CI [0.04, 0.27]; $p = .009$) and in the right entorhinal region (d = 0.13, 95% CI [0.13, 0.16]; $p = .03$), SA was never higher (Figs E and F in S1 Appendix). When comparing AN-BP and AN-R, no differences were revealed after FDR correction for multiple comparisons.

### z-scores from CentileBrain normative model

**AN versus HC.** In individuals with AN, 35/68 CT metrics showed a significantly higher-than-expected proportion of participants with supranormal z-scores based on the normative reference, while 23/68 showed a significantly higher-than-expected proportion of participants with infranormal z-scores (Fig 2a, 2b). The corresponding proportions for SA were 17/68 for supranormal values, and 0/68 for infranormal z-scores (Fig 2c). The corresponding proportions for SV were 0/14 for supranormal z-scores and 13/14 for infranormal z-scores (Fig 2d). The inclusion of global measures (e.g., total volume

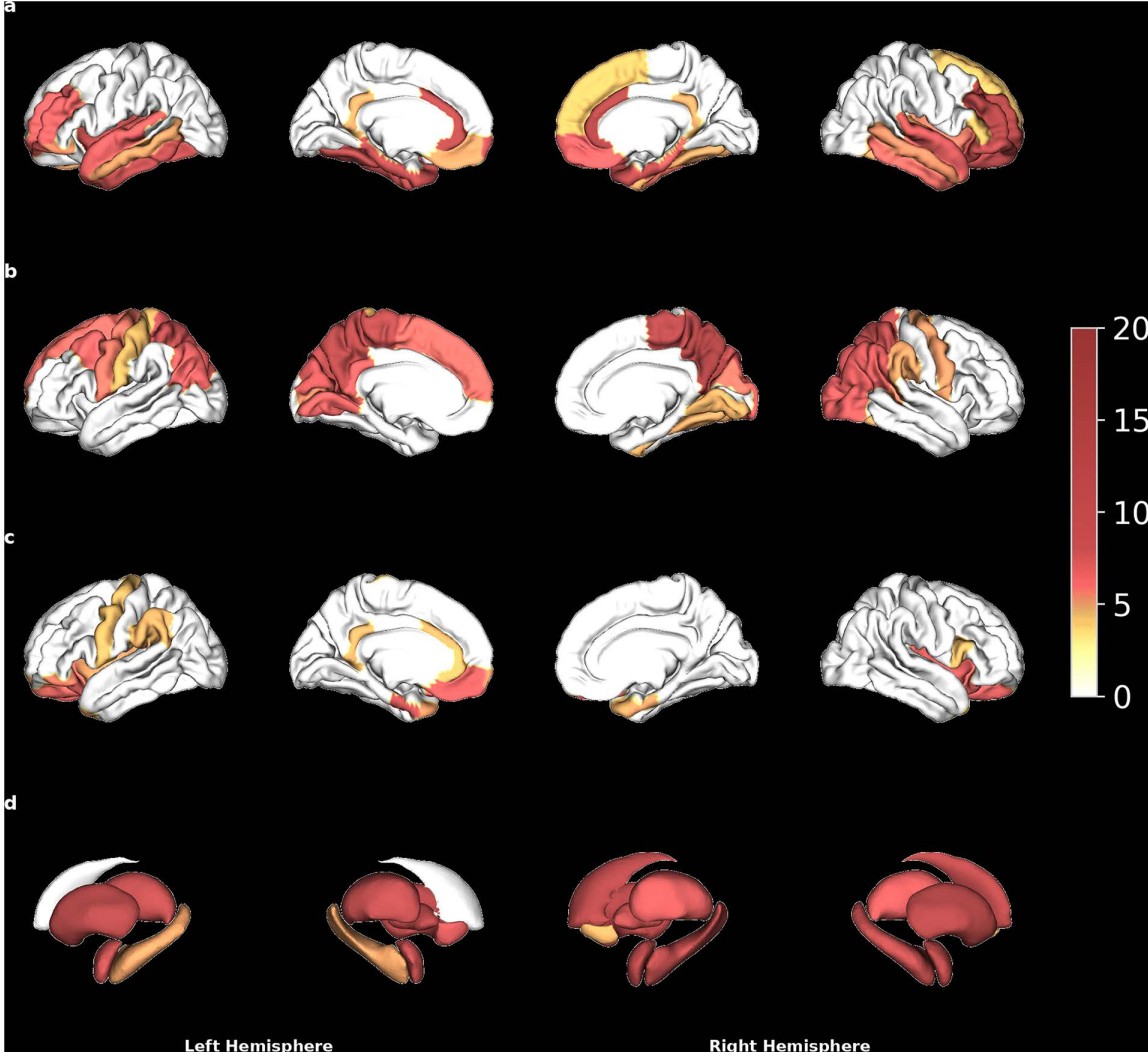

**Fig 2. Percentage of individuals in the anorexia nervosa (AN) group with extreme z-scores in regional metrics of brain structure, based on the normative model from the CentileBrain Project (an initiative of the ENIGMA (Enhancing NeuroImaging Genetics through Meta-Analysis) Lifespan Working Group).** Colored regions indicate areas where the percentage of extreme z-scores was significantly higher than expected in the normative reference after a false discovery rate (FDR) correction for multiple comparisons, with red hues representing higher percentages than yellow hues. Shown are **(a)** percentages of supranormal z-scores for cortical thickness (CT) metrics (threshold: 4.04%), **(b)** percentages of infranormal z-scores for CT metrics (threshold: 4.21%), **(c)** percentages of supranormal z-scores for cortical surface area (SA) metrics (threshold: 4.21%), and **(d)** percentages of infranormal z-scores for subcortical volume (SV) metrics (threshold: 4.39%), see also Section A.6 in S1 Appendix. For no regional metrics of SA the percentage of infranormal z-scores and for no regional metrics of SV the percentage of supranormal z-scores were significantly higher than expected in the normative reference after FDR correction for multiple comparisons.

of GM) as explanatory variables in the normative model warrants caution in the interpretation of these results. Specifically, a supranormal deviation in CT within a given region may not indicate a higher absolute CT value in that region, since the CentileBrain model controls for mean CT. For example, this might explain why not in all regions with a mean CT reduction in AN (Fig 1a), a higher-than-expected proportion of patients with infranormal z-scores was found (Fig 2b). This becomes evident when comparing raw individual measures of brain structure with centile curves illustrating age dependence (Figs G and H in S1 Appendix).

Group-level heterogeneity indices were elevated across CT, SA, and SV for AN compared to HC (all $p < .001$, see Section B.1 in S1 Appendix).

**AN-R versus AN-BP.** No differences in the proportions of infranormal or supranormal z-scores between AN-BP and AN-R were significant in any of the brain morphology phenotypes considered (Table E in S1 Appendix). Overlaps and nonsignificant differences between subtypes in the metrics where higher proportions of infranormal and supranormal z-scores were observed are reported in Section B.2 and displayed in Fig I in S1 Appendix.

### Machine learning classification

Classification of AN from HC achieved high, above chance performance (ROC-AUC[LSsO] = 0.75, PR-AUC[LSsO] = 0.75, $p < .001$; ROC-AUC[ComBat-GAM] = 0.81, PR-AUC[ComBat-GAM] = 0.80, $p < .001$), see Fig J in S1 Appendix. For context, ROC–AUC values of 0.5 reflect chance-level performance, with values around 0.7–0.8 commonly considered acceptable discrimination and values ≥0.8 indicating strong discrimination, while PR–AUC values should be interpreted relative to the class prevalence, with values exceeding the corresponding baseline (~0.44) reflecting meaningful precision–recall performance. Proportion of deviance not explained by confounding variables was above chance ($p < .001$, Table G in S1 Appendix). The most relevant features were CT metrics in the parietal cortex and thalamus volume, both showing significant negative importance, indicating that lower values contributed to classification in the AN group (Fig 3). In contrast, CT in the temporal pole and entorhinal cortex had positive importance (albeit 4–5 times smaller in magnitude than the negative importances found for CT metrics in the parietal cortex), suggesting that higher CT values supported classification in the AN group (Fig 3).

In contrast, classification of AN-BP from AN-R did not achieve a performance above chance (ROC-AUC[LSsO] = 0.54, PR-AUC[LSsO] = 0.28, $p > .14$; ROC-AUC[ComBat-GAM] = 0.55, PR-AUC[ComBat-GAM] = 0.30, $p > .12$), with PR-AUC values close to the baseline defined by class prevalence (~0.24), see Fig K in S1 Appendix. Employing a neural network instead of SVM did not change these results (Section A.6, Figs L and M in S1 Appendix).

### Discussion

In this large-scale multisite mega-analysis, we confirmed robust and widespread structural GM reductions (90% of CT, 84% of SA and 100% of SV metrics affected) in individuals with AN compared to HC, consistent with previous meta-analytic findings [6]. The strongest effects were observed for CT (mean Cohen's $d = -0.54$), followed by SV (mean $d = -0.37$), and SA (mean $d = -0.23$), underscoring the pronounced and spatially extensive nature of brain structure abnormalities in AN. Using a normative modeling approach (CentileBrain), we further showed that individuals with AN had a significantly higher-than-expected proportion of extreme (infra- or supra-normal) z-scores in regional brain metrics, particularly for CT. Specifically, z-scores for each participant were computed relative to predictions from the normative model for a reference female of the same age and, importantly, matched global structural properties (mean CT, total SA, and total GM volume). While SA measures showed (relative) localized increases in supranormal values, SV were characterized by a predominance of infranormal deviations across nearly all structures. Finally, machine learning reliably distinguished individuals with AN from HC based on structural MRI metrics (ROC-AUC = 0.75–0.81), indicating that brain morphological features are sufficiently altered to allow for accurate individual-level classification. Critically, however, no significant differences emerged between AN-R and AN-BP, neither in group-level comparisons, nor in the frequency of normative deviations. Even machine learning methods, which are sensitive to complex multivariate patterns, failed to classify AN subtypes above chance level. These findings suggest that brain structural differences between AN-R and AN-BP are subtle or nonexistent.

The results for the univariate group comparisons closely aligned with those from our previous meta-analysis, which was based on a larger number of sites [6], and other previous studies [40,53]. For CT, the agreement on regions with a significant difference with our previous prospective meta-analysis [6] was 95.59% (in the remaining 4.41% regions, the

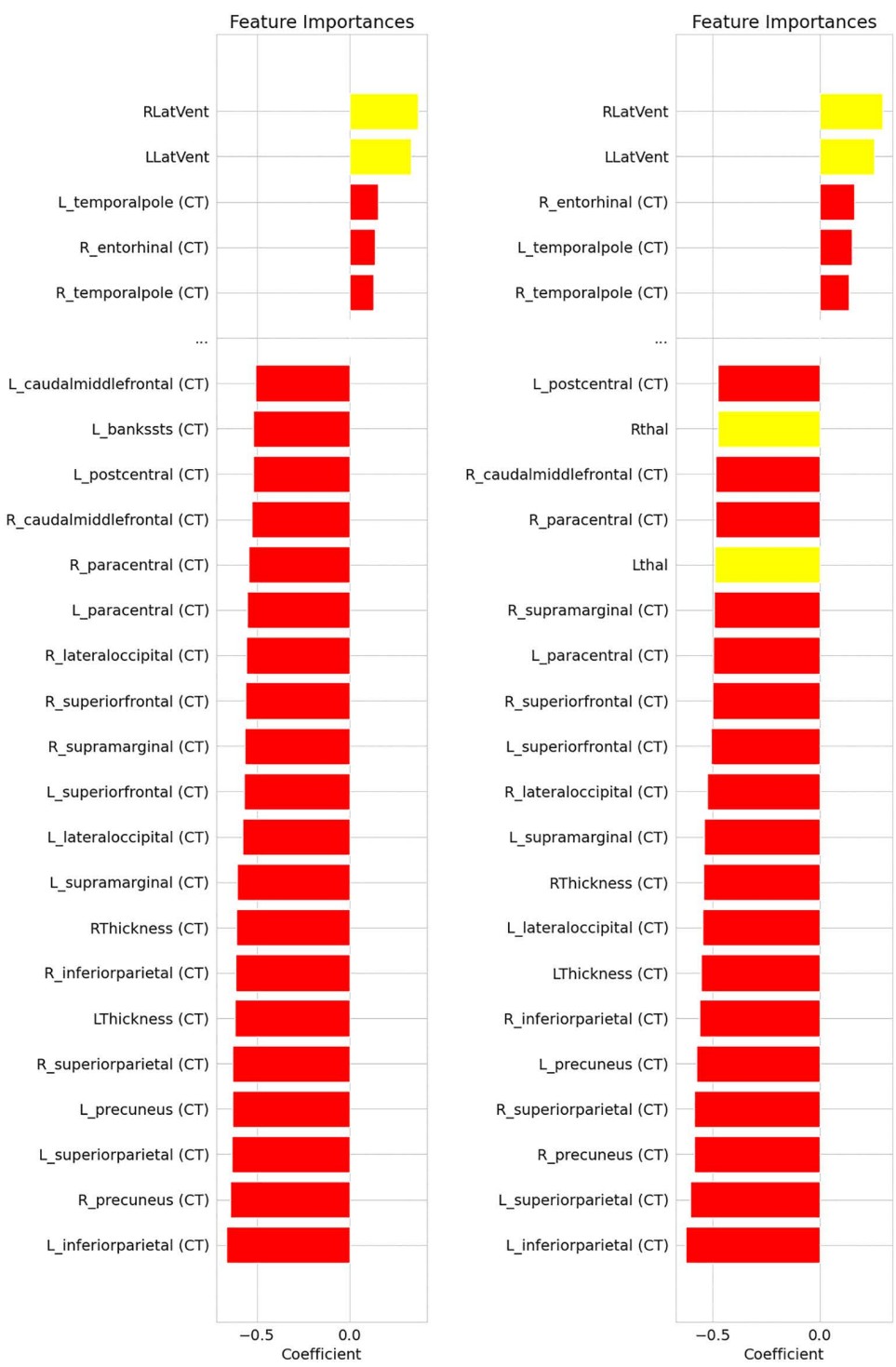

**Fig 3. Feature importances for the anorexia nervosa (AN) vs. healthy controls (HC) classification.** To assess the importance of each feature for the classification, we computed the Pearson correlation coefficient between each feature and the machine learning-based risk score as in Haufe and colleagues [50]. More positive/negative values indicate that a larger/smaller value for a feature supports classification in the AN group. We display the 5/20 features with the largest positive/negative importance, for the ComBat with generalized additive models (ComBat-GAM, left) and Leave-Site(s)-Out cross-validation (LSsO, right) pipelines. Only features whose importance was significant after false discovery rate (FDR) correction for multiple comparisons are listed. Feature importances for metrics of cortical thickness (CT) are shown in red, while metrics of subcortical volume (SV) and cerebrospinal

fluid spaces are shown in yellow. Metrics of cortical surface area (SA) were not among the most important features. L=left hemisphere; R=right hemisphere; Thal=thalamus; Caud=caudate nucleus; Put=putamen; Pall=pallidum; Hipp=hippocampus; Amyg=amygdala; Accumb=nucleus accumbens; LatVent=lateral ventricles.

mega-analysis detected a difference but not the meta-analysis), for SA 72.06% (in 22.06%/5.88% regions the mega-/meta-analysis detected a difference but not the meta-/mega-analysis), and for SV the agreement was 100%. These findings suggest that the current mega-analytic approach provided greater statistical power, enabling the detection of significant effects even for metrics with smaller effect sizes. Specifically, for CT metrics where a significant reduction was also found in Walton and colleagues [6], the effect sizes were comparable (d=−0.55(0.21) versus d=−0.57(0.20) in Walton and colleagues [6]), while in the left rostral anterior cingulate only our analysis detected an effect (d=−0.11). Importantly, in contrast to Walton and colleagues [6] and reviews that reported either cortical thinning or no significant differences—but not increases—in AN [53–56], we observed higher CT in the left temporal pole and the right entorhinal cortex in underweight patients with AN. This finding is compatible with large single-site studies measuring CT vertexwise [40,41], and a machine learning study suggesting that higher CT in these regions helps classify individuals as having AN [5]. Notably, these regions show little to no normative cortical thinning during adolescence [57,58], suggesting that they follow distinct maturational trajectories that may confer resilience to starvation-induced cortical loss. Together with the observation that the effect sizes for CT reductions span a range between d=−0.11 and d=−0.96, this finding suggests that, while cortical thinning and overall reductions in GM volume in AN are widespread and predominantly driven by BMI [40,41], some regions are more susceptible to structural changes, potentially reflecting region-specific cellular composition or metabolic vulnerability [40]. Despite widespread alterations likely dominated by starvation-related metabolic vulnerability, it is informative to consider how the regions showing the largest or most distinct effects relate to AN traits or phenotypes. The strongest CT reductions were observed in posterior parietal regions, implicated in visuospatial processing and mental imagery and repeatedly discussed in relation to altered body representation, body image distortion, and multisensory integration in AN [59,60]. For SA, the most pronounced effects were located in the bilateral transverse temporal gyri, classically associated with early auditory processing [61]; links to core AN phenotypes are necessarily speculative here and may reflect more general, distributed state-related effects rather than a specific auditory phenotype. For SV, the largest reductions involved the bilateral thalamus, a central hub for sensory relay and cortico–subcortical integration [62]. Thalamic volume reductions have been consistently reported in AN and linked to metabolic and endocrine markers, including leptin levels and BMI-related pathways ([63]. Finally, a small number of regions showed opposite-direction effects, with higher CT in the left temporal pole and right entorhinal cortex. Given the modest effect sizes and the multiple-comparisons context, these findings should be interpreted cautiously. Nonetheless, these medial and anterior temporal regions are implicated in socio-emotional [64] and memory-related processes [65] and may reflect inter-individual heterogeneity, compensatory mechanisms, or trait-like differences in subsets of individuals rather than starvation effects per se. The results for the frequencies of extreme z-scores as assessed within the normative model from the CentileBrain group [36] need to be interpreted with care. Since z-scores were adjusted for global structural metrics, a z-score outside the normative range does not imply an absolute increase or decrease in a given region, but rather a deviation relative to normative individuals with similar global brain characteristics. For CT, both supranormal (primarily in temporal and frontal regions) and infranormal (mainly in parietal, occipital, and frontal regions) z-scores were more frequent in AN than expected, indicating significant disruptions of the normative profile beyond the global thinning pattern in a significantly high number of patients. For SA, only the frequency of supranormal deviations was significantly elevated, while for SV, there was a consistent and widespread increase in the frequency of infranormal z-scores. This suggests a more heterogeneous pattern of deviations in the AN group for brain morphology metrics. Indeed, on average, standard deviations of individual z-scores were higher in the AN group compared to the normative reference for both CT, SA, and SV metrics. This finding suggests that AN is not associated with a uniform shift in brain structure, but rather with genuine biological heterogeneity: some patients show

severe alterations in a given metric, while others are minimally affected. Normative-modelling studies across multiple psychiatric disorders have reported similar findings: group averages provide limited information, and individuals show heterogeneous deviations from typical brain organization [32,35,66].

Machine learning classification well above chance indicates that brain scans from individual participants with AN can be reliably distinguished from those of HC. The superior performance of the ComBat-GAM compared to the LSsO pipeline, together with the absence of overfitting, supports its effectiveness in mitigating site effects. This suggests that the model learned to rely on biologically meaningful structural MRI features rather than technical artefacts. In contrast, the lower performance with LSsO suggests that uncorrected site-specific variability can hinder generalization. Notably, while the ComBat-GAM setup enables testing on unseen data from known sites, the LSsO framework provides a more stringent evaluation by estimating generalization to entirely new sites, where site effects cannot be adjusted for in advance. In the ComBat-GAM pipeline, the most relevant features were CT metrics in the frontal, parietal, and occipital lobes, all showing significant negative importance—indicating that lower values contributed to classification in the AN group. For the LSsO pipeline, in addition to CT metrics, also the thalamus volume had higher negative importance. In contrast, CT in the temporal pole and entorhinal cortex had positive importance for both pipelines (albeit 4–5 times smaller in magnitude than the negative importances found for CT metrics in the parietal cortex), suggesting that higher values supported classification in the AN group. These results were consistent with the univariate results, which found larger effect sizes for CT reductions in parietal regions and higher CT in the AN group in the entorhinal cortex and temporal pole. Similarly, the frequency of infranormal z-scores for CT in parietal regions was significantly elevated in AN compared to the normative reference. SA metrics were not among the most important features for classification, and compatibly, the reductions in univariate analyses had smaller effect sizes. SV had somewhat in-between negative effect sizes in univariate analyses, and the frequence of infranormal z-scores was elevated for nearly all metrics, but SV features contributed less to classification compared to CT metrics. While the aim of brain morphology–based AN versus HC classification was not to propose a substitute for BMI in routine diagnosis, using brain measures might become relevant in situations where differential diagnosis is challenging, such as atypical presentations [67], apparent absence of other AN criteria due to poor insight or treatment ambivalence [68], or comorbid medical conditions [69,70]. For this reason, we believe that future machine-learning studies would benefit from explicitly including underweight HC or individuals with low BMI due to nonpsychiatric causes, as this would allow a more direct assessment of the incremental value of brain morphology beyond BMI. However, assembling such samples requires careful clinical characterization to exclude eating disorders and to account for somatic causes of low weight [71].

Compared to other multicentric studies in psychiatric populations, the classification performance achieved here (ROC-AUC[LSsO] = 0.75, ROC-AUC[ComBat-GAM] = 0.81) was notably higher than that reported by the ENIGMA Bipolar Disorder group [48] using similar cross-validation methods (ROC-AUC[LSsO] = 0.61), and also exceeded performance in major depressive disorder, where accuracies below 60% were reported using brain morphology data [72], but was comparable to results reported for schizophrenia [73]. Except for Arold and colleagues [5], previous machine learning studies in AN were based on single-site data with fewer than 50 participants per group, yet also reported high classification accuracy [74–78]. Overall, these findings suggest that brain alterations in underweight individuals with AN can be detected at the individual level, despite substantial variability—both across imaging sites (e.g., scanner differences and protocol variations) and across individuals (e.g., genetic and environmental factors).

However, whether such alterations differ meaningfully between clinical subtypes of AN remains unclear. In this respect, this study investigated structural brain differences between the restricting (AN-R) and binge-eating/purging (AN-BP) subtypes of AN using three complementary analysis approaches. After controlling for BMI, no significant differences emerged between subtypes in metrics of CT, SA, or SV—neither in conventional univariate analyses nor in the frequency of supranormal and infranormal z-scores derived from the CentileBrain normative model [36]. Furthermore, it was not possible to classify individuals from these subgroups using a machine learning algorithm. These findings suggest that AN-R and AN-BP may share core neurobiological features that are not distinguishable using structural MRI. While the clinical

distinction between these subtypes remains relevant—given documented differences in emotion regulation [79], treatment outcome [14], and medical complications [80–84]—frequent transitions between subtypes, particularly from AN-R to AN-BP and often accompanied by clinical worsening [11] have led some researchers to propose that AN-R may represent a transitional phase rather than a stable subtype [13,19,51,52]. From this perspective, the absence of marked structural brain differences observed in this study may reflect the temporal fluidity of subtype expression, such that individuals classified as AN-R and AN-BP do not constitute biologically distinct groups but rather represent different points along a shared disease continuum. To better capture the biological heterogeneity within AN, future studies should consider data-driven approaches—such as unsupervised machine learning—to identify subgroups that transcend traditional symptom-based classifications and may better reflect treatment outcomes or long-term illness trajectories [85].

These results should be interpreted in light of several limitations. First, we did not assess or control for psychiatric comorbidities (e.g., obsessive-compulsive disorder, depression). However, previous large-scale studies have reported relatively small effects of these conditions on brain structure [24,86,87], making it unlikely that our findings in the AN versus HC comparison are primarily driven by comorbidities. Second, the present design does not allow a definitive disambiguation between neurobiological differences related to AN and effects related to starvation or low BMI, as BMI was not included as a covariate in the primary AN versus HC analyses due to its strong confounding with diagnostic group membership. In this respect, a recent study comparing underweight HC to individuals with AN found both alterations related to underweight and alterations specific to AN [7]. Further insights may come from studies including individuals recovered from anorexia nervosa, in whom the effects of acute starvation are expected to be minimal [40,41,88]. Third, because the ethnoracial/ancestry composition of the contributing cohorts could not be harmonized and is likely skewed toward participants of European ancestry, the generalizability of our findings to other ethnoracial groups remains uncertain. Fourth, we compared two relatively simple classification algorithms—support vector machines and shallow neural networks—for distinguishing individuals with AN-R from AN-BP and observed no better-than-chance performance. It remains possible that more advanced models, larger datasets, or the use of raw neuroimaging data as input could yield different results. Finally, distinctions between AN subtypes might emerge more clearly by investigating recovery dynamics longitudinally or when employing other imaging modalities, such as diffusion tensor imaging or functional MRI, which may capture dimensions of brain function or connectivity not reflected in structural MRI.

We confirmed large and widespread reductions in GM in AN, most pronounced in CT and SV, while SA was less affected. Analysis of extreme deviations from the normative reference indicated a higher frequency of extreme z-scores, and a higher degree of heterogeneity in AN. In line with this, machine learning models successfully distinguished individuals with AN from HCs, with CT metrics contributing more strongly to classification than SV metrics—even though univariate analyses also revealed significant SV reductions. No significant differences between AN subtypes were found in any structural metric or in the frequency of extreme deviations, and machine learning classifiers could not distinguish between AN-R and AN-BP above chance level. To better understand heterogeneity in AN, future studies should apply data-driven techniques—such as unsupervised machine learning—to identify biologically meaningful subgroups or psychopathological dimensions linked to treatment outcomes.

## Supporting information

**S1 Appendix. Contains Supplementary Methods, Supplementary Results, Supplementary References, Supplementary Tables A–G, and Supplementary Figs A–M.**
(DOCX)

**S1 STROBE Checklist.** STROBE checklist for case-control studies. Checklist reproduced from the STROBE Statement (https://www.strobe-statement.org/; von Elm and colleagues, PLoS Med. 2007;4(10):e296. https://doi.org/10.1371/journal.pmed.0040296) under the Creative Commons Attribution 4.0 International License (CC BY 4.0).
(DOC)

## Author contributions

**Conceptualization:** Fabio Bernardoni, Stefan Ehrlich.

**Data curation:** Fabio Bernardoni, Dominic Arold, Klaas Bahnsen.

**Formal analysis:** Fabio Bernardoni.

**Funding acquisition:** Fabio Bernardoni, Stefan Ehrlich.

**Investigation:** Fabio Bernardoni, Dominic Arold, Klaas Bahnsen, Lasse Bang, Federico D'Agata, Giovanni Abbate-Daga, Christian K. Tamnes, Iain Campbell, Owen O'Daly, Ulrike Schmidt, Guido Frank, Stefanie Horndasch, Andreas Hess, Arnd Dörfler, Hans-Christoph Friederich, Joe Simon, Angela Favaro, Luca Lavagnino, Christina E. Wierenga, Amanda Bischoff-Grethe, Amy E. Miles, Allan Kaplan, Aristotle Voineskos, Paul A. M. Smeets, Annemarie A van Elburg, Unna Danner, Joseph A King.

**Methodology:** Fabio Bernardoni, Dominic Arold, Ruiyang Ge, Clara Moreau, Sophia I. Thomopoulos, Neda Jahanshad, Sophia Frangou.

**Project administration:** Fabio Bernardoni, Laura Berner, Stefan Ehrlich.

**Resources:** Fabio Bernardoni, Stefan Ehrlich.

**Software:** Fabio Bernardoni, Dominic Arold, Luis Schoppik, Ruiyang Ge.

**Supervision:** Sophia Frangou, Joseph A. King, Paul Thompson, Stefan Ehrlich.

**Visualization:** Fabio Bernardoni.

**Writing – original draft:** Fabio Bernardoni, Stefan Ehrlich.

**Writing – review & editing:** Dominic Arold, Luis Schoppik, Klaas Bahnsen, Ruiyang Ge, Clara Moreau, Lasse Bang, Federico D'Agata, Giovanni Abbate-Daga, Christian K. Tamnes, Iain Campbell, Owen O'Daly, Ulrike Schmidt, Guido Frank, Stefanie Horndasch, Andreas Hess, Arnd Dörfler, Hans-Christoph Friederich, Joe Simon, Angela Favaro, Luca Lavagnino, Christina E. Wierenga, Amanda Bischoff-Grethe, Amy E. Miles, Allan Kaplan, Aristotle Voineskos, Paul A. M. Smeets, Annemarie A. van Elburg, Unna Danner, Sophia I. Thomopoulos, Laura Berner, Neda Jahanshad, Sophia Frangou, Joseph A. King, Paul Thompson.

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
