## [Editor Report · Decision Letter 0]

29 Oct 2025

Dear Dr bernardoni,

Thank you for submitting your manuscript entitled "Brain morphology in Anorexia Nervosa and its subtypes: Mega-Analysis, Normative Modelling, and Machine Learning Approaches" for consideration by PLOS Medicine.

Your manuscript has now been evaluated by the PLOS Medicine editorial staff, and I am writing to let you know that we would like to send your submission out for external peer review.

For clinical studies, please upload a copy of your trial study protocol as a supporting information file. The study protocol should be the version submitted for approval to the institutional review board or ethics committee, should include any amendments to the study protocol, as well as the date of their approval by the institutional review or ethics committee. Please also detail any deviations from the study protocol in the Methods section of your manuscript. The editors will consider the protocol and study conduct prior to a final decision for external review.

Please re-submit your manuscript within two working days, i.e. by Oct 31 2025 11:59PM.

Kind regards,

Heather Van Epps, PhD

Consulting Editor

PLOS Medicine

---

## [Decision Letter · Decision Letter 1]

9 Jan 2026

Dear Dr bernardoni,

Many thanks for submitting your manuscript "Brain morphology in Anorexia Nervosa and its subtypes: Mega-Analysis, Normative Modelling, and Machine Learning Approaches" (PMEDICINE-D-25-03733R1) to PLOS Medicine. The paper has been reviewed by subject experts and a statistician; their comments are included below and can also be accessed here: [LINK]

As you will see, the reviewers found this work to be well-conducted and of potential clinical relevance. Further clarifications and explanations, especially regarding the relationship of brain morphology with the BMI, have been required and are listed in detail at the end of this letter. After discussing the paper with the editorial team and an academic editor with relevant expertise, I'm pleased to invite you to revise the paper in response to the reviewers' comments. We plan to send the revised paper to some or all of the original reviewers, and we cannot provide any guarantees at this stage regarding publication. Please be advised that we may seek the input of an additional independent reviewer to consider a revised manuscript.

We ask that you submit your revision by Jan 28 2026 11:59PM. However, if this deadline is not feasible, please contact me by email, and we can discuss a suitable alternative.

Don't hesitate to contact me directly with any questions (efourli@plos.org).

Best regards,

Evangelia

Evangelia Fourli,

Associate Editor

PLOS Medicine

efourli@plos.org

Comments from the editorial team:

Please revise the abstract to address the following points:

- Intro: please explain more clearly how this study addresses the issue identified in the final sentence of the Background section. This explanation should be incorporated as the new concluding sentence of the Background portion.

- Methods and Findings: The key novel results could be more clearly described in the Methods and Findings section of the abstract. Please revise accordingly.

Comments from the reviewers:

Reviewer #1: "Brain morphology in Anorexia Nervosa and its subtypes: Mega-Analysis, Normative Modelling, and Machine Learning Approaches" analyzed data from 11 international sites in the ENIGMA Eating Disorders Working Group, in particular cortical thickness and subcortical volumes in anorexia nervosa (AN) subjects versus healthy controls (HC). Three modelling techniques (statistical mega-analysis, normative modelling, machine learning) were applied. In general, they all found significant (and discriminative) differences between the AN and HC groups with respect to various quantitative measures. This may enable the classification of AN subjects with improved accuracy as compared to MRI-based psychiatric classification.

While the presented evidence is largely convincing, some issues might be considered:

1. In the Introduction section, it is stated that a previous prospective harmonized meta-analysis from the ENIGMA Eating Disorders (ED) Working Group revealed sizeable and widespread gray matter (GM) reductions associated with low weight and body mass index (BMI) in AN. It might be briefly clarified whether these correlations were likewise observed with naturally lower weight and BMI subjects (that did not have AN), if possible.

2. In the Study samples subsection, the female-only exclusion criteria might be briefly substantiated, and the diagnosis of AN (as opposed to HC) briefly described.

3. In the Case-control sample (AN vs. HC) subsection, the selection of cases and any associated matching of AN to HC cases, might be briefly described.

4. In the Group Comparisons section, it is stated that univariate mega-analyses were performed. It might be clarified as to whether any covariates/confounds (e.g. age, weight) were considered and/or applied to the analyses, or if these were assumed to be addressed by the ComBat-GAM harmonization process (as detailed in Supplementary 1.5)

5. In the Normative modeling section, the CentileBrain brain morphometry models are stated to be sex-specific, but derived from an ethnoracial diverse sample of >37,000 individuals aged 5-90 years. It might be briefly clarified as to the extent to which this reference population coincides with that for the collated ENIGMA EDWG data, specially given the relatively lower ages of the study population (from Table S1 etc.)

6. For Normative modeling, results for the HC group would be relevant too, and might be presented if possible.

7. In the Classification Pipelines subsection, it is stated that ComBat-GAM and all preprocessing steps were applied in a cross-validated manner. It might be clarified whether this cross-validation concurs with the ten-fold cross-validation (with grid search) later discussed in the Hyperparameters optimization and performance estimation section, and extends also to the training and test folds.

8. In Supplementary 1.8, further details on grid values and results appear to be in Table S4, and not S3.

9. In general, a pertinent question would be whether discrimination of AN and HC via brain morphology metrics offers any advantages over that of simply using BMI/weight, which would be a relatively more-accessible modality. This might be briefly discussed, possibly together with a simple analysis on the classification performance of BMI towards AN/HC.

Reviewer #2: I thank the author team for an fascinating read on such a comprehensive mega-analysis evaluating morphological differences in those with anorexia nervosa and their subtypes. I not only commend the wide range of methodological approaches implemented in this study, but also the extensive array of available data utilised to address research questions. All feedback and suggestions may be found within the attached documentation.

Introduction

1. Minor: the following sentence requires a hyphen for ‘group level’, as well as a space between ‘subtypes’ and ‘[7]’: “In contrast to our previous group level meta-analysis, analyzing individual-level data enables examination of variability related to clinical AN subtypes[7].” Introduction P.4

2. To assist prospective readers, I’m wondering if the authors may be able to further elaborate on the distinction between the AN-BP subtype and bulimia nervosa. Recent literature reviewing AN subtypes has debated the classification and operationalisation of AN-BP and AN-R subtypes (Albracht-Schulte KD, Flynn L, Gary A, Perry CM, Robert-McComb JJ. The Physiology of Anorexia Nervosa and Bulimia Nervosa. InThe Active Female: Health Issues throughout the Lifespan 2023 Feb 26 (pp. 95-117). Cham: Springer International Publishing.), as well as significant diagnostic crossover and transition between subtypes (Serra R, Di Nicolantonio C, Di Febo R, De Crescenzo F, Vanderlinden J, Vrieze E, Bruffaerts R, Loriedo C, Pasquini M, Tarsitani L. The transition from restrictive anorexia nervosa to binging and purging: a systematic review and meta-analysis. Eating and Weight Disorders-Studies on Anorexia, Bulimia and Obesity. 2022 Apr;27(3):857-65.).

I think this debate is important for the author team to acknowledge, and has additional implications on study hypotheses associated with structural morphology of the brain. Introduction P.4

3. Similar to the comment above, I think it is pertinent to touch upon previous literature aiming to investigate brain morphological or functional distinctions between AN subtypes (e.g., Yonezawa H, Otagaki Y, Miyake Y, Okamoto Y, Yamawaki S. No differences are seen in the regional cerebral blood flow in the restricting type of anorexia nervosa compared with the binge eating/purging type. Psychiatry and clinical neurosciences. 2008 Feb;62(1):26-33.), and how the availability and richness of data present in this study may be more suitable to comprehensively investigate these nuances.

I note that the authors have begun to touch upon this at the bottom of page four within their introduction, but I think that discussion of previous literature would further strengthen the study rationale.

Introduction P.4

4. Minor: There is an issue with plurality in the following text: “(e.g., cortical thickness (CT) in specific brain region)” Introduction P.5

Methods

1. Minor: As the authors have already abbreviated ‘the ENIGMA Eating Disorder Working Group’ to read as ‘The ENIGMA ED working group’ within the Introduction, I would suggest they remain consistent in use of the abbreviation within the methods and subsequent text. Methods – Study samples P.6

2. Could the authors elaborate on the BMI cut-offs used for adults within their inclusion and exclusion criteria? ICD-11 classifications characterise AN as having a BMI falling below 18.5 kg/m2, while DSM-5 classifications include BMI ranges, which mark AN severity as mild (BMI ≥ 17 kg/m2), moderate (BMI 16–16.99 kg/m2), severe (BMI 15–15.99 kg/m2) or extreme (BMI < 15 kg/m2; Himmerich H, Treasure J. Anorexia nervosa: diagnostic, therapeutic, and risk biomarkers in clinical practice. Trends in Molecular Medicine. 2024 Apr 1;30(4):350-60.).

Considering the present range of <17.5 for AN falls between these classifications, it would benefit readers for the authors to rationale this cut-off. This includes elaborating upon including adults with a BMI below a healthy range within their HC group. I hope the authors have a solidified rationale for this decision, considering a significant portion of neuroanatomical research into AN has failed to accurately distinguish brain atrophy associated with malnutrition from AN-specific morphological changes. Methods – Study samples P.6

3. I think the reader would benefit from a short, sentence-long description as to how AN subtypes were classified prior to referencing the Supplementary Methods.

Methods – Sample of cases with subtype information (AN-R vs. AN-BP) P.6

4. Which MRI manufacturers and models were utilised for T1-weighted image acquisition across sites? Retrospectively, I can see that these are provided within Table S3, but can the authors make mention of this within the metnhods?

Methods – Image acquisition and processing P.7

5. Minor: The reference for Pomponio et al. (2020) is not numbered, and references Bahnsen et al. 2022 and Bernardoni et al. 2016 seem to use a different referencing style than the rest of the manuscript. Methods – Group comparisons P.7

6. Minor: Should CentileBrain be appropriately referenced at first mention? Methods – Normative modeling P.8

7. Could the authors include mention of using 2-proportion z-tests to assess proportions of infranormal or supranormal z-scores between AN-BP and AN-R? Methods – Normative modeling P.8

8. Minor: Arold et al. (2023) should be appropriately numbered to remain consistent with referencing style. Methods – Machine learning classification P.8

9. Minor: Under ‘Explainable AI’, “AI” should be spelled out in full at first mention Methods – Machine learning classification P.10

Results

1. Minor: the AN subtypes should be abbreviated in the following sentence for consistency: “The age difference in subtypes may reflect the more typical transition from the restricting to the binge-eating/purging subtype over time [14, 43, 44].” Results - Demographics P.10

2. Minor: Please remain consistent with spelling of “Mean (SD)”; at times it is presented with a space before brackets (e.g., ‘Mean (SD)’), and during other times no space is present (e.g., ‘Mean(SD)’). Results – Univariate comparisons P.10

3. To provide readers with a bit of context, could the authors elaborate on what would generally be considered appropriate or high performance for classifier models, particularly in reference to ROC-AUC values? I think the provision of referential values would ease reader interpretation of these findings, particularly those less well-versed in machine learning techniques. Results – Machine learning classification P.14

Discussion

1. Minor: Is it necessary to re-introduce the abbreviations for cortical thickness, subcortical volume and cortical surface area within this section?

Discussion P.16

2. Minor: Walton et al. (2022) should be appropriately numbered, and the year should be included within the following portion of text: “the effect sizes were comparable (d=-0.55(0.21) vs. d=-0.57(0.20) in Walton et al.), while […]” Discussion P.16 & P.17

3. Is there extant literature that the authors are able to draw on regarding their interpretation of findings stemming from their analysis of z-scores from the CentileBrain group? Have other studies utilised this approach, and can the authors explicitly acknowledge this? The authors are making multiple interpretations in this section that, without reference to adequate evidence, borders on speculation rather than interpretation. Discussion P.17

4. The following text should fall within the methods or Supplementary Methods section, rather than within the discussion: “To address site-related variability, we implemented two complementary strategies. In the first, we applied ComBat-GAM to explicitly remove site effects prior to model training [33]. In the other, site correction was not performed; instead, we used a leave-site-out cross-validation (LSsO) strategy, ensuring that all test data came from sites unseen during training.” Discussion P.17

Figures/Tables

1. The following text should be included within the methods or Supplementary Methods, rather than within the figure legend: “To assess the importance of each feature for the classification, we computed the Pearson correlation coefficient between each feature and the machine learning-based risk score (Haufe et al., 2014)”. As a minor point, the reference should also be appropriately numbered as per the recommended style. Figure 3. P.15

Supplementary Methods

1. The authors note here that participants were classified as having AN with a BMI <18.5, but this value is noted as 17.5 within the main manuscript. Could the authors clarify? 1.2 Study Criteria P.29

2. Table S3-Table S7 are missing appropriate abbreviations sections. Table S3-Table S7 P.45-P.48

3. Similar to the supplementary tables, Figure S1-Figure S13 inconsistently display footnotes to clarify included abbreviations. Figure S1-Figure S13 P.49-P.61

Reviewer #4: Dear Authors and Editor,

Thanks for letting me read this interesting and well-written paper!

I need to directly state that I am not well experienced with imaging and the analyses conducted here, thus my review will be based on the more general parts of the paper and its design.

Major comments:

If I understand it correctly BMI was only taken into account in the subtype analyses, for which no differences were detected, and thus the findings could all be related to starvation/low BMI. Have the authors tested what happens when controling for BMI in the main analyses, i.e. AN vs HC? In line with this I also lack inclusion/sub-analyses of individuals recovered from AN, and if the number of studies including recovered patients are not enough (likely) for a meta-/megaanalyses this needs to be discussed and mentioned as a limitation.

I lack a discussion and speculation on how the specific regions in which differences are detected could relate to AN traits and phenotypes.

Minor comments:

Different BMI criteria are stated in main text vs supplementary text, <17.5 vs <18.5. Or does the later only apply to patients in recovery phase? Please clarify.

I question the statement on p. 4t: "... and more severe medical complications due to the cycle of binge-eating and

purging."

---

* Please upload any figures associated with your paper as individual TIF or EPS files with 300dpi resolution at resubmission; please read our figure guidelines for more information on our requirements: http://journals.plos.org/plosmedicine/s/figures. While revising your submission, we strongly recommend that you use PLOS's NAAS tool (https://ngplosjournals.pagemajik.ai/artanalysis) to test your figure files. NAAS can convert your figure files to the TIFF file type and meet basic requirements (such as print size, resolution), or provide you with a report on issues that do not meet our requirements and that NAAS cannot fix.

After uploading your figures to PLOS's NAAS tool - https://ngplosjournals.pagemajik.ai/artanalysis, NAAS will process the files provided and display the results in the "Uploaded Files" section of the page as the processing is complete.

If the uploaded figures meet our requirements (or NAAS is able to fix the files to meet our requirements), the figure will be marked as "fixed" above. If NAAS is unable to fix the files, a red "failed" label will appear above.

When NAAS has confirmed that the figure files meet our requirements, please download the file via the download option, and include these NAAS processed figure files when submitting your revised manuscript.

* Please provide URLS for all funders under the Financial Disclosure.

FIGURES AND TABLES

SUPPLEMENTARY MATERIAL

REFERENCES

OBSERVATIONAL STUDIES

* Abstract: Please include the study design, population and setting, number of participants, years during which the study took place (enrollment and follow up), length of follow up, and main outcome measures.

* Please ensure that the study is reported according to the STROBE (or appropriate STOBE extension) guideline (available from: https://www.equator-network.org/reporting-guidelines/strobe) and include the completed STROBE (or STROBE extension) checklist as Supporting Information. Please add the following statement, or similar, to the Methods: "This study is reported as per the Strengthening the Reporting of Observational Studies in Epidemiology (STROBE) guideline (S1 Checklist)." When completing the checklist, please use section and paragraph numbers, rather than page numbers.

* [FOR POPULATION HEALTH/REGISTRY STUDIES] Please ensure that the study is reported according to the RECORD guideline (available from https://www.record-statement.org) and include the completed checklist as Supporting Information. Please add the following statement, or similar, to the Methods: "This study is reported as per the Reporting of Studies Conducted using Observational Routinely-Collected Data (RECORD) guideline (S1 Checklist)." When completing the checklist, please use section and paragraph numbers, rather than page numbers.

* [FOR POPULATION HEALTH ESTIMATES] Please ensure that the study is reported according to the GATHER statement (available from https://www.equator-network.org/reporting-guidelines/gather-statement) and include the completed checklist as Supporting Information. Please add the following statement, or similar, to the Methods: "This study is reported as per the Guidelines for Accurate and Transparent Health Estimates Reporting (GATHER) statement (S1 Checklist)." When completing the checklist, please use section and paragraph numbers, rather than page numbers.

* [FOR MEDIATION ANALYSES] We recommend that the study is reported according to the AGReMA statement (https://agrema-statement.org/#:~:text=AGReMA%20is%20an%20evidence%2D%20and,randomised%20trials%20and%20observational%20studies) and include the completed checklist as Supporting Information. Please add the following statement, or similar, to the Methods: "This study is reported as per the Guideline for Reporting Mediation Analyses (AGReMA) statement (S1 Checklist)." When completing the checklist, please use section and paragraph numbers, rather than page numbers.

* For all observational studies, in the manuscript text, please indicate: (1) the specific hypotheses you intended to test, (2) the analytical methods by which you planned to test them, (3) the analyses you actually performed, and (4) when reported analyses differ from those that were planned, transparent explanations for differences that affect the reliability of the study's results. If a reported analysis was performed based on an interesting but unanticipated pattern in the data, please be clear that the analysis was data driven.

* Please state in the Methods section whether the study had a prospective protocol or analysis plan. If a prospective analysis plan (from your funding proposal, IRB or other ethics committee submission, study protocol, or other planning document written before analyzing the data) was used in designing the study, please include the relevant document(s) with your revised manuscript as a Supporting Information file to be published alongside your study and cite it in the Methods section. A legend for this file should be included at the end of your manuscript. If no such document exists, please make sure that the Methods section transparently describes when analyses were planned, and when/why any data-driven changes to analyses took place. Changes in the analysis, including those made in response to peer review comments, should be identified as such in the Methods section of the paper, with rationale.

MODELLING STUDIES

The following list is derived from Geoffrey P Garnett, Simon Cousens, Timothy B Hallett, Richard Steketee, Neff Walker. Mathematical models in the evaluation of health programmes. (2011) Lancet DOI:10.1016/S0140-6736(10)61505-X:

* If pertinent, please provide a diagram that shows the model structure, including how the natural history of the disease is represented, the process and determinants of disease acquisition, and how the putative intervention could affect the system.

* Please provide a complete list of model parameters, including clear and precise descriptions of the meaning of each parameter, together with the values or ranges for each, with justification or the primary source cited and important caveats about the use of these values noted.

* Please provide a clear statement about how the model was fitted to the data, including goodness-of-fit measure, the numerical algorithm used, which parameter varied, constraints imposed on parameter values, and starting conditions.

* For uncertainty analyses, please state the sources of uncertainties quantified and not quantified [can include parameter, data, and model structure].

* Please provide sensitivity analyses to identify which parameter values are most important in the model. Uncertainty estimates seek to derive a range of credible results on the basis of an exploration of the range of reasonable parameter values. The choice of method should be presented and justified.

* Please discuss the scientific rationale for the choice of model structure and identify points where this choice could influence conclusions drawn. Please also describe the strength of the scientific basis underlying the key model assumptions.

* For studies that develop a prediction model or evaluate its performance, please ensure that the study is reported according to the TRIPOD statement (https://www.equator-network.org/reporting-guidelines/tripod-statement) and include the completed checklist as Supporting Information. Please add the following statement, or similar, to the Methods: "This study is reported as per the Transparent Reporting of a Multivariable Prediction Model for Individual Prognosis Or Diagnosis (TRIPOD) statement (S1 Checklist)." For studies using machine learning, please use the TRIPOD-AI checklist. When completing the checklist, please use section and paragraph numbers, rather than page numbers.

---

## [Decision Letter · Decision Letter 2]

14 Apr 2026

Dear Dr. bernardoni,

Thank you very much for re-submitting your manuscript "Brain morphology in Anorexia Nervosa and its subtypes: Mega-Analysis, Normative Modelling, and Machine Learning Approaches" (PMEDICINE-D-25-03733R2) for review by PLOS Medicine.

I have discussed the paper with my colleagues and the academic editor and it was also seen again by 4 reviewers. I am pleased to say that provided the remaining editorial and production issues are dealt with we are planning to accept the paper for publication in the journal.

[LINK]

We look forward to receiving the revised manuscript by Apr 20 2026 11:59PM.

Sincerely,

Evangelia Fourli, Ph.D.

Senior Editor

PLOS Medicine

plosmedicine.org

Requests from Editors:

GENERAL EDITORIAL REQUESTS

Please note that some of the following requests may not apply to your manuscript.

"* At this stage, we ask that you include a short, non-technical Author Summary of your research to make findings accessible to a wide audience that includes both scientists and non-scientists. The Author Summary should immediately follow the Abstract in your revised manuscript. This text is subject to editorial change and should be distinct from the scientific abstract. Ideally each sub-heading should contain 2-3 single sentence, concise bullet points containing the most salient points from your study. In the final bullet point of ‘What Do These Findings Mean?’ Please include the main limitations of the study in non-technical language.

Please see our author guidelines for more information: https://journals.plos.org/plosmedicine/s/revising-your-manuscript#loc-author-summary."

* Please confirm that your title complies with PLOS Medicine's style. Your title must be nondeclarative and not a question. It should begin with main concept if possible. "Effect of" should be used only if causality can be inferred, i.e., for an RCT. Please place the study design ("A randomized controlled trial," "A retrospective study," "A modelling study," etc.) in the subtitle (ie, after a colon).

I would strongly advise you to modify the title of your study to adhere to PLOS title guidelines. As an example you can instead use: "Brain morphology in Anorexia Nervosa and its subtypes: A multi-cohort study of individual participant data". This suggestion is not intended to be prescriptive, and you are encouraged to revise it as you see fit.

* Please confirm that your abstract complies with our requirements, including format (three sections: Background, Methods and Findings, and Conclusions) and providing all the information relevant to this study type https://journals.plos.org/plosmedicine/s/submission-guidelines#loc-abstract

* Please ensure that the Introduction ends with a clear description of the study question or hypothesis.

* Please ensure that all abbreviations are defined at first use throughout the text.

* Please confirm that all numbers presented in the abstract are present and identical to numbers presented in the main manuscript text.

* Please ensure that the study is reported according to the STROBE guideline, and include the completed STROBE checklist as Supporting Information. Please add the following statement, or similar, to the Methods: "This study is reported as per the Strengthening the Reporting of Observational Studies in Epidemiology (STROBE) guideline (S1 Checklist)."

When completing the checklist, please use section and paragraph numbers, rather than page numbers."

While STROBE is recommended for observational studies, authors may additionally follow or substitute other relevant reporting guidelines where more appropriate to the specific study design or analytical approach, provided this is clearly stated and justified.

GENERAL

* Please review your text for claims of novelty or primacy (e.g. 'for the first time') and remove this language. In addition, please check that any use of statistical terms (such as trend or significant) are supported by the data, and if not please remove them.

* In the abstract, please include the important dependent variables that are adjusted for in the analyses.

* Please remove the 'conclusions' subheading from the discussion. Please also remove any other subheadings from the discussion.

"* Statistical reporting: Please revise throughout the manuscript, including tables and figures.

- Please report statistical information as follows to improve clarity for the reader ""22% (95% CI [13,28]; p</=)"".

- Please separate upper and lower bounds with commas instead of hyphens as the latter can be confused with reporting of negative values.

- Please repeat statistical definitions (HR, CI etc.) for each set of parentheses."

* In the abstract, please include the important dependent variables that are adjusted for in the analyses.

FUNDING STATEMENT

* The funding statement should include: specific grant numbers, initials of authors who received each award, URLs to sponsors’ websites. Also, please state whether any sponsors or funders (other than the named authors) played any role in study design, data collection and analysis, the decision to publish, or preparation of the manuscript. If they had no role in the research, include this sentence: “The funders had no role in study design, data collection and analysis, decision to publish, or preparation of the manuscript.”

*Please add the relevant URL for each funder.

COMPETING INTERESTS STATEMENT

* All authors must declare their relevant competing interests per the PLOS policy, which can be seen here: https://journals.plos.org/plosmedicine/s/competing-interests For authors with ties to industry, please indicate whether any of the interests has a financial stake in the results of the current study.

DATA AVAILABILITY

"* PLOS Medicine requires that the de-identified data underlying the specific results in a published article be made available, without restrictions on access, in a public repository or as Supporting Information at the time of article publication, provided it is legal and ethical to do so. Please see the policy at

http://journals.plos.org/plosmedicine/s/data-availability

and FAQs at

http://journals.plos.org/plosmedicine/s/data-availability#loc-faqs-for-data-policy "

"* The Data Availability Statement (DAS) requires revision. For each data source used in your study:

"

*Please provide third party contact information for data access; authors or the principal investigator cannot be used as the contact person.

* Thank you for agreeing to make your data available. At this time, please provide the link to the data repository and accession numbers required for access. Please check that the link you provided is correct as I cannot access it. Please also archive the code and provide the relevant link, eg via Zenodo.

ETHICS AND CONSENT

* Please provide the name(s) of the institutional review board(s) that provided ethical approval.

* Please, include a paragraph in the main methods where you clearly state what type of consent was obtained and a short description of your ethics approval, which then cites the supplementary material for further information.

FIGURES

* Please define all elements of box plots in the figure caption - center line, box limits and whiskers.

* Please provide titles and legends for all figures and tables (including those in Supporting Information files). Please define all acronyms used in each figure or table in its corresponding legend.

* Please convert any pie charts to another data representation for example a table, or other type of graph.

* Please show graph axes beginning at zero. If this is not possible, please show a break in the axis.

* Please consider avoiding the use of red and green in order to make your figure more accessible

Acknowledgments

*Please separate the Acknowledgements and Disclosures section into two individual sections.

Comments from Reviewers:

Reviewer #1: We thank the authors for addressing our previous concerns.

Reviewer #2: Manuscript Title: Brain morphology in Anorexia Nervosa and its subtypes: Mega-Analysis, Normative Modelling, and Machine Learning Approaches

The following revised manuscript provides a mega-analysis of neuroimaging data from 11 international sites to investigate structural differences in those with anorexia nervosa (AN) relative to healthy controls. The authors have comprehensively attended to the points brought up by all reviewers. My previous review of this manuscript comprised of potential concerns regarding the classification of AN subtypes, descriptions of group-based BMI cut-offs and interpretation of CentileBrain-derived z scores, which I believe have been adequately addressed.

I only have minor grammar- or punctuation-based suggestions for this work, as outlined below:

1. Methods (Study samples; P.10; Lines 279-280): Minor - I think there may be a plurality issue in the following sentence and would recommend the following:

"We jointly optimized the model hyperparameters (e.g., number of PCA components) via a stratified (i.e., keeping the group proportions constant across partitions) ten-fold cross-validated grid search (Table S4)"

2. Methods (Hyperparameters optimization and performance estimation; P.11; Lines 295-296): Minor - The following sentence requires a space between the text and bracket:

"To better approximate such scenarios, we also used Leave-Site(s)-Out cross-validation(LSsO, Nunes et al.2018)"

Reviewer #3: The current revised version provided new information in a high-quality way. I have no more questions.

Reviewer #4: All the comments raised by me and the other reviewers have been responded to and taken into account in a satisfying manner.

[LINK]

---

## [Editor Report · Decision Letter 3]

23 Apr 2026

Dear Dr. bernardoni,

Thank you very much for re-submitting your manuscript "Brain morphology in Anorexia Nervosa and its subtypes: A multi-cohort study of individual participant data" (PMEDICINE-D-25-03733R3) for review by PLOS Medicine.

While you have addressed most of the editorial requests, there are still some points that require careful revision. The remaining editorial issues that need to be addressed are listed at the end of this email.

We expect to receive your revised manuscript within 5 days. Please email us (plosmedicine@plos.org) if you have any questions or concerns.

We look forward to receiving the revised manuscript by Apr 28 2026 11:59PM.

Sincerely,

Evangelia Fourli, Ph.D.

Senior Editor

PLOS Medicine

plosmedicine.org

Requests from Editors:

* Data Availability Statement (DAS). The DAS statement is different between manuscript files and metadata. Please make sure that all metadata exactly match statements/text within the manuscript files. Regarding the DAS statement please use the one from the manuscript files.

* Thank you for updating the OSF link, which is now accessible. Please, archive the code and provide the relevant DOI link as well.

* Figures S2 and S4:Please explain why some p-values = 0. If p-value numbers are too small, use p<0.001. I also suggest adding an asterisk when differences are significant for ease of readability (please define the asterisk in the legend if you proceed with the recommendation).

* Please revise the whole manuscript and define all abbreviations in text, figure legends and tables. Especially for figure legends and tables, content should be self-contained; therefore all abbreviations must be defined. Figures 1 and 2 for example are missing relevant abbreviations (AN, HC, CT, GLM).

* Figure 1: the legend is not detailed enough. Please provide more details to facilitate readers, especially relating to significance and colouring presentation.

* Figure 3 lacks a title.

* The introduction should end with a clear description of the study question and hypothesis. Currently, the introduction ends with a short abstract-like format. Please substitute this (Line 175-201) with a small paragraph clearly stating the question and hypothesis. If you wish to keep the format as is, please make sure to conclude the introduction with a clear description of the study question and hypothesis.

* Suggested changes in the paragraph "demographics" in the methods. It is slightly dense. I suggest using "across sites" instead of "overall", removing "instead" (line 358, since it does not contradict the previous statement) and "only" (line 359).

---

## [Editor Report · Decision Letter 4]

30 Apr 2026

Dear Dr bernardoni,

On behalf of my colleagues and the Academic Editor, Dr Perminder Singh Sachdev, I am pleased to inform you that we have agreed to publish your manuscript "Brain morphology in Anorexia Nervosa and its subtypes: A multi-cohort study of individual participant data" (PMEDICINE-D-25-03733R4) in PLOS Medicine.

PRESS

Sincerely,

Evangelia Fourli, Ph.D.

Associate Editor

PLOS Medicine